# 3D-Printed Encapsulation of Thin-Film Transducers for Reliable Force Measurement in Biomedical Applications

**Raffaele Pertusio and Silvestro Roatta ***

Department of Neuroscience, University of Torino, 10125 Torino, Italy
* Correspondence: silvestro.roatta@unito.it

**Abstract:** In biomedical studies as well as in clinical trials, it is often useful to have a reliable measure of the force exerted by the body (e.g., clenching force at the teeth or pinch force at fingertips) or on the body by external stimuli (e.g., taps to elicit reflexes or local pressure for nociceptive stimulation). Thin-film sensors such as FlexiForce® provide a very handy and versatile solution for these applications, but can be easily damaged and offer poor accuracy and repeatability, being heavily affected by the surface material they come into contact with. The aim of the study is the realization of a 3D-printed housing that completely embeds the sensor, thus providing mechanical protection and increasing the reliability of the measurement. The increasing availability of 3D printers and of printing materials for medical use allows the user to shape the housing according to specific needs, with short developing time and low cost.

**Keywords:** Thin-film sensor; force measurement; 3D printing; teeth clenching

## 1. Introduction

Thin-film force sensors are versatile and can potentially be employed in a wide range of applications. For example, due to their low thickness, they have been adopted for the measurement of the forces and pressures exerted on different body parts [1–5], as well as clenching forces [6–9]. However, the behavior of film sensors is strongly influenced by test conditions, e.g., the area and nature of the contact surfaces, load direction, etc. [10,11]. The difficulty in maintaining the same working conditions during the calibration procedure and during the subsequent measurement sessions may impair the accuracy and reliability of the measurements. Indeed, in a wide range of applications from orthodontics to physiotherapy, sensors can come into contact with soft or hard parts of the human body (skin or teeth), with fluids (e.g., saliva), with concentrated loads (for example, teeth occlusion forces), and unknown tangential forces. On this basis, these sensors would benefit from a shield that protects them from structural damage while, at the same time, providing a stable interface, reducing the effect of changes in the stiffness and texture of contact materials. Here, we provide a possible solution that consists of fully embedding the sensor in a 3D-printed rigid housing using a 3D FDM/FFF printing process. This approach is based on printing the housing around the film sensor, which, in addition to granting constant contacting interface and mechanical protection, also allows for designing complex housing shapes, which can fit different applications and purposes. We hypothesized that the 3D-printed housing would not alter the basic load–response curve of the sensor, but would reduce the dependency of the measurement on the contact surface, thus increasing its reliability. We also aimed at testing the performance of three different materials for the housing: PLA (IRA3D, Cressa, Italy, Polylactic Acid) nGen_Flex (ColorFabb, Belfel, The Netherlands, Polymer Amphora Flex FL6000) and ABS (IRA3D, Cressa, Italy, Acrylonitrile–Butadiene–Styrene). PLA and ABS are the most-used FDM printing materials. PLA is also a bioplastic, so its biocompatibility and its biodegradability make it very interesting for the biomedical sector. nGen_Flex is a more flexible and softer material appropriate for biomedical applications.

## 2. Materials and Methods

### 2.1. Thin-Film Sensors

FlexiForce sensors (Tekscan) were employed in the present study. These are thin-film sensors that use a resistive-based technology such that the application of a force to the active sensing area of the sensor results in a change in the resistance of the sensing element [12]. Every sensor has to be preconditioned before use and the construction of a calibration curve is recommended to account for the individual sensitivity of the transducers [12,13], provided that the same sensor configuration is used for calibration and for the actual measurements. The use of a puck is recommended from the manufacturer and, in clinical tests, its use is practically always necessary [14]. The sensor must therefore be incorporated into a shell with an upper and lower protection puck and 2 double-sided adhesive tape layers to avoid slipping between the sensor and the puck. The external shell should then be sufficiently hard to protect the sensor while still allowing for correct sensor loading. Based on these recommendations the sensors have been embedded in a 3D-printed housing, as described below.

### 2.2. 3D Printing

The housing is 3D-printed around the sensor, following the normal printing process steps. The housing was designed (Rhinoceros 7) with the aim of fixing the sensor in a stable position while ensuring that the force lines of the applied load are conveyed through its sensing area. This latter issue was overcome by the circular cantilevered area (see inferior side of the housing in Figure 1a), matching the inner 80% of the sensing area [12,15]. The effectiveness of this solution was verified with a finite-element model simulation, as reported in Figure 1d–f. A detailed design of the housing is given in Supplementary Figure S1, and a file for 3D printing is freely available at https://cults3d.com/en/3d-model/various/encapsulation-of-thin-force-sensor-flexiforce (accessed on 18 February 2023).

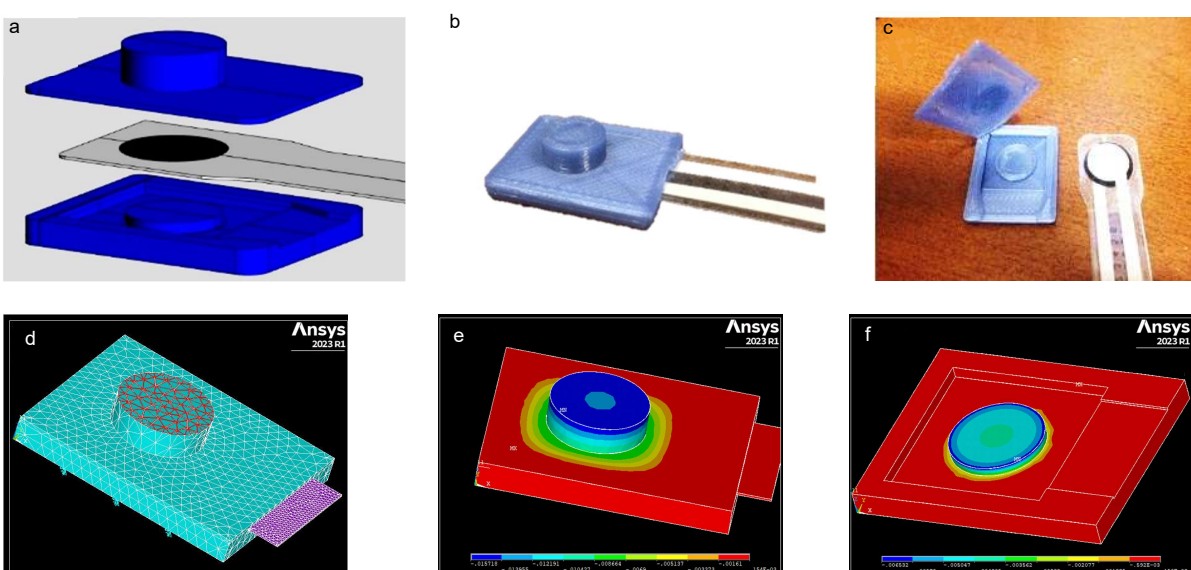

**Figure 1.** (**a**) 3D model of the housing realized with Rhinoceros 7 software. Note how the film sensor (in grey) is positioned over the bottom part of the housing before the upper part is printed on top; (**b**) example of housed sensor; (**c**) upper and lower sides of the housing have been separated with a cutter to show inner side. The material in this example is PLA. At the bottom, a finite-element model simulation of the elastic deformation of the housing exposed to a 400 N load is shown; (**d**) definition of the mesh; (**e**) deformation of the external upper surface; (**f**) deformation of the inner inferior surface (blue = highest; red = lowest). Note that the load is transmitted to the lower layer exclusively through the circular cantilevered area corresponding to the sensing area of the transducer. The FEM model of the housing has been realized with Ansys 2023 R1 software.

The slicer software is IRA3D (Cura). The 3D printer is IRA3D Poetry2 Dual Extrusion (2015). The printing temperature was set to 200° for PLA, 250° for ABS, and 225° for nGen_Flex. Fortunately, they are materials with different printing temperatures, so the full range of normal printing temperatures was used. In this respect, we here point out that the printing process of the ABS housing damaged all 3 sensors, possibly due to the high printing temperature. Thus, no result will be reported for ABS.

Printing velocities were set in the range of 30–40 mm/s, and the thickness layer for all the printing processes was 0.2 mm, yielding a satisfactory printing quality. Printing speed results from a trade-off between printing accuracy and the risk of overheating the sensor: it is preferable to have high speeds and low printing temperatures to avoid damaging the sensor. The cooling action of the fans may help. The printing is automatically stopped upon completion of the lower part of the housing, and the FlexiForce sensor is then inserted and blocked in its position by means of double-sided tape. The double-sided tape, placed on the top face of the sensor, improves adhesion. Therefore, we printed the top of the housing on the sensor. The upper layer of envelope was 0.6 mm (3 layers), which guarantees adequate mechanical protection to the sensor. It is possible to reduce the layers to 2 when the applied loads are small. Making less than 2 layers may not be sufficient to ensure adequate mechanical resistance of the housing. A cylindrical tip (2.75 mm in height) on the upper side further strengthens the structure and concentrates the load on the sensitive area of the FlexiForce. The total thickness of the printed housing was 5.15 mm. The printing time was about 5 min. A photograph of the housings made of the three different materials is presented in Supplementary Figure S2.

### 2.3. Experimental Set-Up

Load–response curves of naked and housed sensors were constructed by means of the set-up depicted in Figure 2. The sensor was connected to a circuit for conditioning and amplification [12,13], and the output was fed into a signal acquisition board (Micro1401, CED, UK) along with the calibrated output of a load cell (YZC-516 50 Kg, GuangCe, Guangzhou, China). Acquired signals were digitally transmitted and stored on a personal computer. A clamp was used to manually generate the load: within the clamp the sensor was positioned "in series" with the load cell, so that both simultaneously shared the same load when the clamp was tightened. A puck was positioned between the sensor and the load cell. Pucks of 3 materials with different stiffnesses (Table 1) were used: Steel, PLA, and Nitrile Butadiene Rubber (NBR).

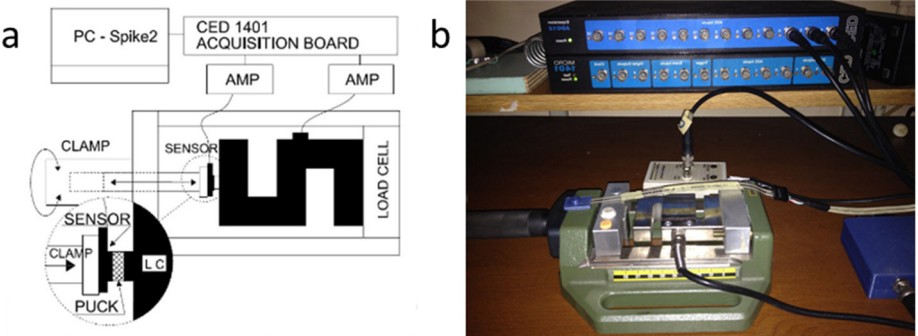

**Figure 2.** Scheme (**a**) and picture (**b**) of the experimental setup. LC: load cell, AMP: signal conditioning and amplification for both the sensor and the load cell.

**Table 1.** Materials used for 3D printing or as puck.

| Acronym | Material | Used for | Young Modulus (MPa) |
|---|---|---|---|
| PLA | Polylactic Acid | 3D Housing/Puck | 2500–3500 |
| nGen_Flex | Polymer Amphora Flex FL6000 | 3D Housing | 100–150 |
| ABS | Acrylonitrile–Butadiene–Styrene | 3D Housing | 1500–2500 |
| Steel | Steel | Puck | 200,000 |
| NBR | Nitrile Butadiene Rubber | Puck | 4 |

*2.4. Experimental Protocol*

Nine FlexiForce film sensors (Tekscan, Norwood, MA, USA, model A201 445N) were employed. The first three sensors (1–3) were housed with PLA, the second three (4–6) with nGen_Flex, and the last three (7–9) with ABS. Each sensor was tested in three subsequent conditions: (i) without the housing (Naked), (ii) after application of the housing (Housing I), and (iii) after removal of the first housing and application of a new housing (Housing II). In each condition, the sensor was loaded through pucks of three different materials (Steel, PLA, NBR). In each of these 9 conditions, the load–response curve was recorded by manually and slowly increasing the load from 0 to 40 kg (about 90% of the full scale) and then decreasing back to 0 kg. Prior to each measurement session, the naked and housed sensors were preconditioned according to the manufacturer's guidelines.

*2.5. Analysis and Statistics*

Hysteresis (%) was calculated as the maximum difference between the loading and unloading curve divided by the maximum value of the output voltage (maximum load applied = 40 kg). Linearity was evaluated with the coefficient of determination $r^2$ for a trend line calculated with the minimum squares and intercept equal to 0 (maximum load applied = 40 kg). Linear mixed model was chosen as statistical method to compare the different working conditions (sensor output at 40 kg) because the study presents both repeated measures and hierarchical structures of the data. The assumption of normality of residuals was verified by the Shapiro–Wilk test. The linear mixed model was used to break down the between-group variance, due to the specific sensitivity of each sensor, from the within-group variance, due to the variations in the contact surface of the puck (steel, PLA, NBR), using the sensor condition as factor grouping (Naked, Housing I, and Housing II). The heteroscedasticities of the between-group and within-group variances have been studied with a log-likelihood test (REML: restricted maximum likelihood [15]) with the parameter AIC as measure of model fit (a lower AIC score is better). Variability of the output due to different pucks was quantified by the coefficient of variation (CoV = SD/mean). The CoV was compared in the three different sensor conditions by means of the Student's t test. Analysis was performed with the statistical package R 4.10.

**3. Results**

Representative examples of a load–response curve of a naked sensor tested with pucks of different materials are reported in Figure 3a. It can be observed that, although the curves are qualitatively similar, the magnitude of the response (i.e., the sensitivity of the sensor) is affected by the puck. In comparison, the housed sensor presents a remarkably similar response irrespective of the puck (Figure 3b). Similar results were obtained for nGen_Flex (Figure 3) and PLA (not shown).

The sensor output at 40 kg loading was used to compare the responses in the different conditions (Naked, Housing I, Housing II) and with different contact materials (Steel, PLA, NBR). The results for the three sensors housed with PLA have shown that the sensor output did not depend on the condition nor on the puck material. As shown in Figure 4a, the output of the sensor is not modified by the housing nor by the puck material. However, it can be observed that the variability of the measurements due to different pucks is higher in the naked conditions compared with the housed conditions. The results are qualitatively

similar for the three sensors housed with nGen_Flex, as shown in Figure 4b. Again, it can be observed that the puck largely affects the variability of the sensor output, but only in the naked condition. It can be observed that in some cases the housing increased the sensitivity of the sensor, with an increase in output voltage at a given load, while in other cases the sensitivity decreased. These differences may be attributed to small differences in the printing process.

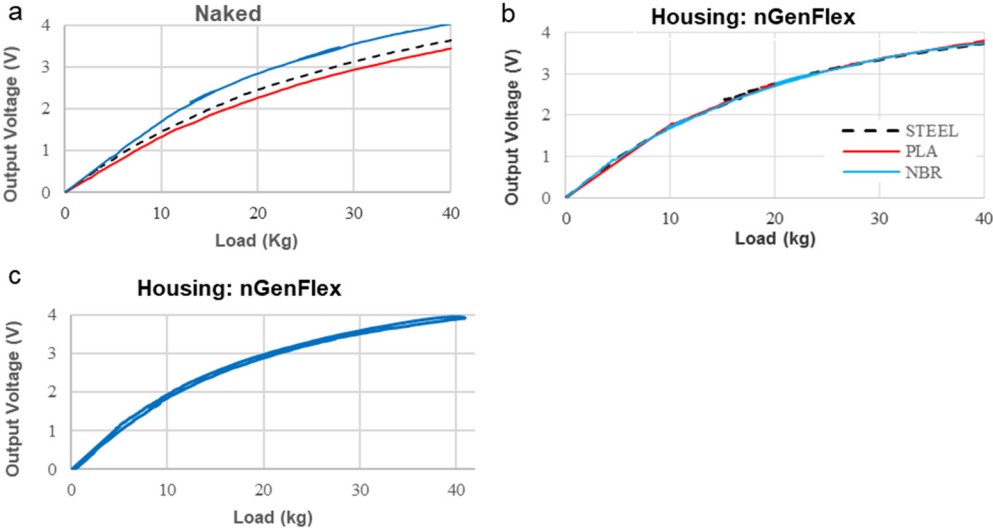

**Figure 3.** (**a**) Load–response curves for naked sensor tested with pucks of different materials. (**b**) Load–response curves for housed sensor (nGen_Flex) tested with pucks of different materials. (**c**) Full-response curve for a housed sensor (nGen_Flex housing) showing the small difference between loading and unloading. For the sake of clarity, only the loading part of the curve is shown in (**a,b**).

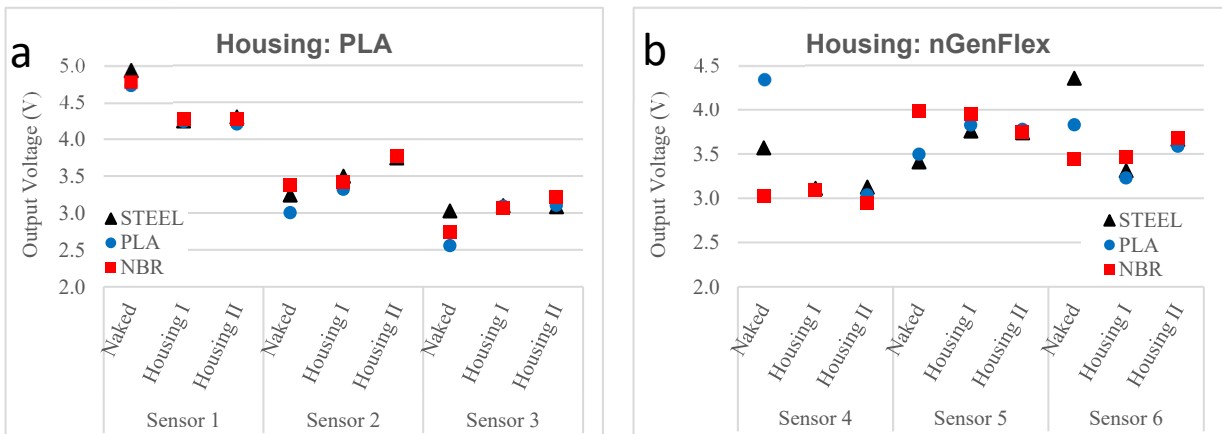

**Figure 4.** (**a**) Response of the sensors to a 40 kg load when tested with pucks of different materials (Steel, PLA, NBR) in the different conditions (Naked, Housing I, Housing II) for housing made of PLA; (**b**) for housing made of nGen_Flex. Note the larger variability of the measurements in the naked condition.

The models (Model Ho-He) with homogeneity of between-variance and heterogeneity of within-variance are significantly better than models (Model Ho-Ho) with homogeneity of between-variance and within-variance (Naked Housing I, Model Ho-Ho to Model Ho-He: AIC 37 to 18, NAKED= 0.104 -HOUSING I = 0.005, *p* value < 0.001; Naked-Housing II, Model Ho-Ho to Model Ho-He: AIC 37 to 12, NAKED = 0.104 -HOUSING II = 0.003, *p* value < 0.001). The models (Model He-He) with heterogeneity of between-variance and within-variance did not lead to significant improvements: the between-variance is

almost the same before and after 3D encapsulation. The printing process does not alter the between-variance and improves the within-variance. This result is further supported by the analysis of the CoV of the measurement.

The larger variability produced by the different pucks in the naked condition compared with the housed conditions is quantified by the coefficient of variation CoV and shown in Figure 5.

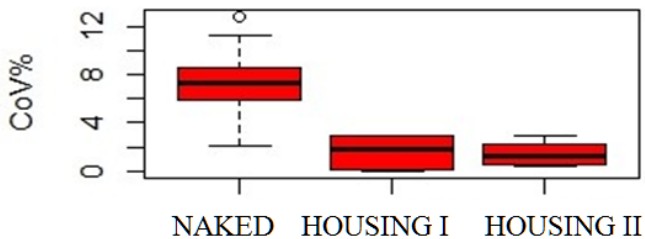

**Figure 5.** Coefficient of variation (CoV) of the responses to 40 kg loading with pucks of different materials in the different sessions. Error bars represent 95% confidence intervals.

In the naked condition (Naked: 8.3%), the CoV is significantly higher than in the housed conditions (Housing I: 1.84%, $p < 0.01$; Housing II: 1,10%, $p < 0.01$). The hysteresis between the loading and unloading curves (see representative example in Figure 3c) was not prominent in naked sensors (4.8% average over sensors 1–6) and was not affected by the housing (Housing I: 4.8%, Housing II: 4.5%). Additionally, the linearity for the housed sensors ($r^2 = 0.90$ average for Housing I, $r^2 = 0.92$ average for Housing II) is similar to the naked sensors ($r^2 = 0.89$ average over sensors 1–6).

## 4. Discussion

The study demonstrated the possibility of completely embedding a thin-film force sensor within a rigid 3D-printed housing through self-made 3D printing. The presence of a housing did not produce significant changes in the load–response curve of the sensor in terms of sensitivity, linearity, or hysteresis, but it consistently reduced the dependence of the sensitivity on contact materials, thus greatly reducing the output variability in repeated measurements. While these sensors offer acceptable repeatability when tested in the same conditions [14], minor alterations in the contact surface in terms of texture and compliance, as can occur even when repositioning the sensor on the same spot (e.g., Figure 3, naked conditions), may profoundly affect the measurement results [10,11]. It can also be observed in Figure 4 that there is no systematic effect of the puck material on the naked sensor output: sometimes PLA gives the highest output, other times it is steel; sometimes the output is higher than in the housed sensor, sometimes it is lower. The dynamic characteristics of the material, such as the elastic return or the viscous components, had no role in the present measurements performed in static conditions. The surface rugosity of the material could play a role, but the three materials were not appreciably different in this respect. Based on these considerations, our conclusion is that the sensor output is extremely dependent on the mechanical interaction between the sensor and the contact surface to the extent that any change in this coupling, including small displacements or sensor repositioning, introduces a large variability in the output. The independence of the force measurement on the contact surface is, however, a crucial need in biomedical applications, where it is not always possible to calibrate sensors in the "in the same conditions" in which they are experimentally employed (as recommended by the manufacturer) [2–5,16]. However, this aspect is often neglected, and the data collected may be highly imprecise. In this respect, 3D-printed housings may provide an easy solution. In fact, the CoV was reduced to less than 2% across measurements on the surfaces of different material. Among the many biocompatible materials on the market, such as PLA and nGen_Flex used in the study, one should opt for those with lower printing temperatures, since too-high temperatures may damage the sensor, as we experienced with ABS. A few issues in the process require

particular attention: the optimal printing temperature to avoid bad adhesion phenomena between layers; proper design of the housing to avoid the development of tangential forces not detectable by the sensor; and adequate stiffness of the housing to avoid deformation of the loading area. Several attempts and prototypes may be necessary before achieving a satisfactory performance of the housed sensor. Fortunately, the relatively low forces (up to 500 N for the sensor adopted in this study) can be well afforded by housings of limited thicknesses, as are necessary for oral application. However, in the case of concentrated forces, such as those transmitted by teeth, it may be appropriate to interpose casts for preventing damage to the housing and better distribute the load over the sensing area. To this purpose, the external surface of the housing could be specifically designed to accommodate paste for dental casting. Alternatively, a rubber cap can serve the purpose of protecting the housing and eliminate the discomfort associated with biting a hard surface: an example of this implementation is shown in Figure 6.

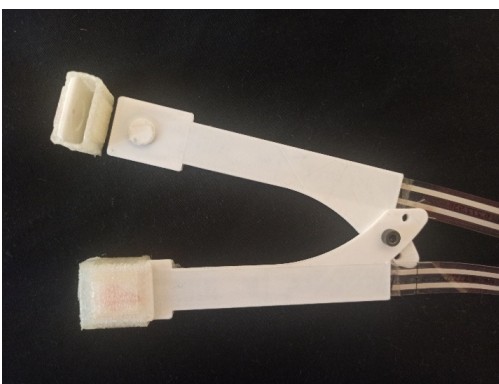

**Figure 6.** Modified implementation of a bilateral bite force measurement, originally proposed by Weisschircher [9]. The sensors are here embedded and sealed within the 3D-printed housing, and 3D-printed rubber cuffs are inserted over the sensing areas.

High-performance printers make it possible to also create very complex models that adapt to precise anatomical parts, thus allowing for easy positioning and use. The sensor housing may also protect the sensor from accidental contact with fluids, such as sweat and saliva, which may provoke irreversible damage. In case the printing process does not grant water proofing, particular care should be followed for sterilization (if required by the application, e.g., by using UV rays rather than liquid agents). The use of UV-C rays of 30 W and wavelengths less than 280 nm 15 min after having thoroughly cleaned the sensor of biological substances (e.g., saliva) with an absorbent cloth and then with cotton soaked in disinfectant (e.g., isopropanol 75%) was suggested by Prusa [17].

Another useful implementation of this methodology concerns the measurement of the exact magnitude and time course of the stimuli delivered with a neurological hammer. The same signal may also serve the purpose of detecting the exact timing of stimulus delivery, as may be necessary to assess the latency of reflex responses. An implementation is presented in Figure 7: the housing embedding the sensor is provided with a handle and a rubber tip, which has to be positioned on the spot to be percussed (e.g., a tendon or a muscle) and then has to be hit by the hammer. In Figure 7c, representative recordings of the patellar reflex are shown. Note the full description of the delivered stimulus, allowing us to estimate a reflex latency of about 39 ms. This device was recently employed for the investigation of tap-induced muscle contractions [1]: pre-positioning the sensor on the tissue rather than sticking it on the hammer was shown to substantially reduce movement artifacts on electromyographic (EMG) signals. In addition, the possibility to record the delivered force stimulus allowed to describe the dependency of the magnitude and spatial extension of the EMG response on the intensity of the stimulus [1].

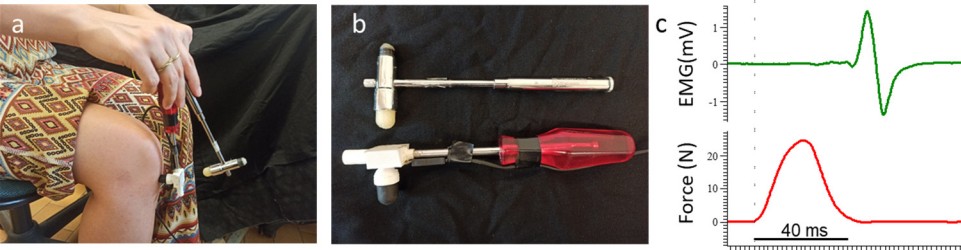

**Figure 7.** Measurement of percussive force stimuli: (**a**) the force sensor is interposed between the patellar tendon and the reflex hammer; (**b**) force sensor (bottom) and reflex hammer (top); (**c**) representative recordings of the patellar reflex: the electromyographic (EMG) response bipolarly recorded from the vastus medialis muscle and the delivered force stimulus (Force).

## 5. Conclusions

The described methodology allows for the easy and cheap implementations of force measurement devices for biological applications, which are based on low-cost thin-film transducers. The embedding of the transducer greatly increases the reliability of the measurement and allows for a comparison of recordings taken from different locations and different subjects. The large variety of materials available for 3D printing, as well as the easiness of customizing the design of the housing, further supports the development of new applications.

**Supplementary Materials:** The following supporting information can be downloaded at: https://www.mdpi.com/article/10.3390/biomechanics3010011/s1, Figure S1: Photograph of the housings made of ABS, nGen_Flex, and PLA, Figure S2: Detailed design of the housing.

**Author Contributions:** Conceptualization, R.P. and S.R.; methodology, R.P.; validation, R.P.; formal analysis, R.P.; resources, S.R.; data curation, R.P.; writing—original draft preparation, R.P.; writing—review and editing, R.P. and S.R.; visualization, R.P.; funding acquisition, S.R. All authors have read and agreed to the published version of the manuscript.

**Funding:** This research received no external funding.

**Institutional Review Board Statement:** Not applicable.

**Informed Consent Statement:** Not applicable.

**Data Availability Statement:** The data presented in this study are openly available in Zenodo at: https://zenodo.org/record/7692161#.ZAGrfx_MK3A, accessed on 25 February 2023.

**Conflicts of Interest:** The authors declare no conflict of interest.

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
