# Peer review of "3D-Printed Encapsulation of Thin-Film Transducers for Reliable Force Measurement in Biomedical Applications"

_2673-7078, doi:10.3390/biomechanics3010011_

Round 1

Reviewer 1 Report

This paper describes a force sensor that is encapsulated in a 3D printed compartment. The encapsulated force sensor can provide high reproducible force measurements regardless of the material of the contact surface. However, in this paper, several important data are missing.

Therefore, the reviewer suggests publishing this paper after several revisions as below.

1.        The authors should provide detail design of the housing. For example, although the authors mentioned that “the force lines of the applied load are conveyed through its sensing area.”, did you confirm it by FEM simulation?

2.        The author should show photographs of three housings (PLA, eGen Flex, and ABS) in SI.

3.        In naked sensors, why is the order of magnitude of sensor outputs not material dependent? For example, in sensor 4, PLA is the highest. On the other hand, in sensor 5, NBR is the highest.

4.        In the measurement of percussive force stimuli, the authors should provide the force data recorded in the demonstration and should discuss the result.

Author Response

This paper describes a force sensor that is encapsulated in a 3D printed compartment. The encapsulated force sensor can provide high reproducible force measurements regardless of the material of the contact surface. However, in this paper, several important data are missing.

Therefore, the reviewer suggests publishing this paper after several revisions as below.

ANSWER: we thank the reviewer for the constructive criticism

  1. The authors should provide detail design of the housing. For example, although the authors mentioned that “the force lines of the applied load are conveyed through its sensing area.”, did you confirm it by FEM simulation?

ANSWER: We thank the reviewer for this suggestion. A FEM simulation was carried-out and the results have been added to Fig.1, demonstrating the effectiveness of the proposed solution.

Detailed design of the housing is now provided in Supplementary Figure 1 and a link to a freely available 3D-printing file was also reported in the text under “2.2. 3D printing”: “The effectiveness of this solution was verified with a finite-element model simulation, as reported in Fig1 (d, e, f). A detailed design of the housing is given in Supplementary Figure 1, and a file for 3D printing is freely available at https://cults3d.com/en/3d-model/various/encapsulation-of-thin-force-sensor-flexiforce.”

  1. The author should show photographs of three housings (PLA, eGen Flex, and ABS) in SI.

ANSWER: An additional figure was uploaded as Supplementary Figure 1

  1. In naked sensors, why is the order of magnitude of sensor outputs not material dependent? For example, in sensor 4, PLA is the highest. On the other hand, in sensor 5, NBR is the highest.

ANSWER: Thank you for raising this issue that was only superficially addressed in the previous version of the manuscript. The following text has been added to the discussion: “It can also be observed in Fig. 4, that there is no systematic effect of the puck material on the naked sensor output: sometimes PLA gives the highest output, other times it is steel; sometimes the output is higher than in the housed sensor, sometimes it is lower. Dynamic characteristics of the material, such as the elastic return or the viscous components had no role in the present measurements performed in static conditions. The surface rugosity of the material could play a role but the three material were not appreciably different in this respect. Based on these considerations, our conclusion is that the sensor output is extremely dependent on the mechanical interaction between the sensor and the contact surface to the extent that any change in this coupling, including small displacements or sensor repositioning, introduce a larga variability in the output”

  1. In the measurement of percussive force stimuli, the authors should provide the force data recorded in the demonstration and should discuss the result.

ANSWER: we are happy to provide original recordings of percussive force stimuli relative to the application presented Fig. 7. A new graph was added showing the force signal of the percussion delivered to the patellar tendon along with the recorded EMG tracing (patellar reflex) recorded from the vastus medialis muscle.

Reviewer 2 Report

The authors provide a materialization of a 3D-printed sensor platform. The idea and design of the experiments were well organized. However, the data and figures can be improved to solidify their conclusion. The detailed issues are mentioned below.

1. The data for ABS as a housing material was not shown due to damage to sensors. This issue can be mentioned in 2.3 Experimental set-up section

2. The actual experimental setup images(picture) should be shown in the manuscript to improve this paper's originality.

3. In load-response curves, the unloading data should be added to demonstrate the physical behaviors of a sensor.

4. The image of Figure 3b should be improved(Lettering of "Housing").

5.  The image of Figure 4 should be improved(Legends for data not shown totally).

6. Please, explain why the output voltage for "naked" samples was higher than those for housing samples.

7. The behavior of the "naked" samples was different according to housing materials(PLA vs. nGen Flex). Please explain these results. 

Author Response

The authors provide a materialization of a 3D-printed sensor platform. The idea and design of the experiments were well organized. However, the data and figures can be improved to solidify their conclusion. The detailed issues are mentioned below.

ANSWER: we thank the reviewer for the constructive criticism

  1. The data for ABS as a housing material was not shown due to damage to sensors. This issue can be mentioned in 2.3 Experimental set-up section

ANSWER: OK, we anticipated that sentence, but we considered the section “2.2. 3D printing” more appropriate, as the printing temperature issue is described there.

  1. The actual experimental setup images (picture) should be shown in the manuscript to improve this paper's originality.

ANSWER OK. A picture of the set-up was inserted beside the scheme, in Fig. 2

  1. In load-response curves, the unloading data should be added to demonstrate the physical behaviours of a sensor.

ANSWER: OK. A representative recording of the full loading-unloading curve was added (Fig. 3c)

  1. The image of Figure 3b should be improved (Lettering of "Housing").

ANSWER: Thank you for noticing this problem. The picture size was enlarged to better accommodate the lowest writings.

  1. The image of Figure 4 should be improved (Legends for data not shown totally).

ANSWER: this was corrected, thank you for noticing it.

  1. Please, explain why the output voltage for "naked" samples was higher than those for housing samples.

ANSWER: Actually, the output of the naked sensor was not systematically higher but, in general, it was just more variable. As requested also by Reviewer 1, this issue is now better discussed and interpreted:  “It can also be observed in Fig. 4, that there is no systematic effect of the puck material on the naked sensor output: sometimes PLA gives the highest output, other times it is steel; sometimes the output is higher than in the housed sensor, sometimes it is lower. Dynamic characteristics of the material, such as the elastic return or the viscous components had no role in the present measurements performed in static conditions. The surface rugosity of the material could play a role but the three material were not appreciably different in this respect. Based on these considerations, our conclusion is that the sensor output is extremely dependent on the mechanical interaction between the sensor and the contact surface to the extent that any change in this coupling, including small displacements or sensor repositioning, introduce a large variability in the output”

  1. The behavior of the "naked" samples was different according to housing materials (PLA vs. nGen Flex). Please explain these results. 

            ANSWER: correct. Accounted for in the answer above.

Round 2

Reviewer 1 Report

The authors completely responsed to the reviewer's comments and appropriately revised the manuscript.